# Gender, status, and team interaction: A microdynamic exploration of wearable sensor data across 11 research groups

Jörg Müller[1]*, Alvaro Uzaheta[2], Julián Salas Piñón[3]

1 Interdisciplinary Research Centre on Social and Cultural Transformations, Universitat Oberta de Catalunya, Barcelona, Spain, 2 Social Networks Lab, ETH Zürich, Zürich, Switzerland, 3 Department of Information and Communications Engineering, Universitat Autònoma de Barcelona, Barcelona, Spain

* jmuller@uoc.edu

## Abstract

Diversity research is increasingly moving beyond a static focus on linear relationships between team-level diversity attributes and outcomes toward a dynamic, configurational perspective on team processes. Recent developments emphasise the structural dimensions of interpersonal relations within teams and how dyadic relationships, mutual perceptions, and behaviours shape team-level outcomes over time. Drawing on real-world organisational data, we apply a hierarchical Dynamic Actor Network Model to examine the role of gender homophily, professional status, and gender-based status cues in face-to-face interactions across 11 R&D teams. Our analysis of wearable sensor data reveals nuanced patterns that challenge any clear-cut diversity effects within teams. Although mixed-gender interactions are generally more common, the interplay between professional status and gender-based status cues changes across organisational contexts. Professional status shows no clear effect on interaction frequency, whereas gender-based status effects are observed in research laboratories but not in private companies. These findings underscore the continued relevance of demographic attributes and associated status dynamics, while highlighting the value of a configurational and temporally sensitive approach to understanding team interaction.

## Introduction

After decades of research on the consequences of diversity, there is a general agreement that the relationship between team diversity and team performance is inconclusive; assumptions on a simple linear effect of diversity characteristics on team performance are only moderate at best and have largely been abandoned [1–3]. This has spurred developments to embrace more complex approaches to understanding the consequences of diversity in work groups. The special edition in the Academy of Management Review, for example, sees "Diversity at a Critical Juncture" [4].

**Data availability statement:** The data underlying the results presented in the study are available from Zenodo under the following URL: https://zenodo.org/records/10968303 The complete set of scripts used for obtaining the results of this study will be published on OSF after acceptance of the article.

**Funding:** This work has received funding from the European Union's Horizon 2020 research and innovaton programme under grant agreement No 665851.

**Competing interests:** The authors have declared that no competing interests exist.

Drawing inspiration from advances in the study of "microdynamics" [5] and "emergent phenomena" in working groups [6,7], diversity research is moving away from static, collectivist accounts of team processes that emphasise simple, linear effects between team-level diversity predictors and outcomes. Instead, current developments focus more on the evolving configurations of interpersonal relations within teams and how these shifting dyadic ties shape team-level outcomes over time [8].

Efforts to move beyond static, collectivist accounts in team research typically involve two methodological steps. The first is to broaden the dominant approach to data collection in diversity research—which has relied heavily on self-reported, questionnaire-based measures—by incorporating richer behavioural data. Although behavioural indicators such as interaction frequency, gaze and body posture vary depending on team members' socio-demographic attributes like gender or race [9–12], most diversity studies continue to rely on cognitive, self-reported measures. This is not surprising, given the considerably higher efforts required for gathering, handling and coding behavioural data [13]. However, the predominant reliance on self-reported data puts the "cognitive cart before the interaction horse" [14] and seriously curbs our capacity to understand the core reality of small groups, namely how face-to-face interaction affects team outcomes [7]. To address this limitation, the following article analyses team interaction data collected with wearable sensors. Wearable sensors offer exciting new possibilities for exploring high-resolution data of previously hard-to-observe behavioural patterns among team members such as entrained body movements, interactions or turn taking [15–17]. Although organisational scholars have addressed the validity and reliability of wearable sensor data [18–20], their uptake in real-world settings remains an exception, limiting our capacities to overcome the inconsistent results of past diversity research [21].

Second, the static collectivist accounts of team diversity remain limited by their tendency to conceive of diversity effects as inherently additive, that is, as "a linear combination of individual team members' attributes" [22]. Rather than concentrating on how cognitive, motivational or demographic measures average across team members, scholars need a more fine-grained assessment of how perceptions of team members change according to the relative positions of actors within the network of interpersonal relations [23]. While there are numerous studies that apply a configurational approach to diversity in teams [24], including the study of team conflict [25,26], or deference in teams [27], empirical studies utilising face-to-face interaction data in this domain remain rare [28].

To address both gaps in the diversity literature, we propose a configurational approach to analyse behavioural interaction data on the role of gender and status effects in teams. Using wearable sensor data to record team interactions, we demonstrate how interactions are conditioned by varying dyadic gender and status configurations among team members. The focus on gender and status is thereby of particular interest, as the intersection between these central diversity concepts in teams remains insufficiently understood [29,30]. Studying helping behaviour in teams, for example, Hong, Lee, and Son [31] argue that dyadic gender composition in relation to status cues plays out in a more complex way than previously considered. As

work group hierarchies in organisations become flatter and less formal, informal status hierarchies become more important as coordination mechanisms that enhance (or undermine) learning at the individual-, group- or organisational level [32,33]. Yet, how gender and status effects can be distinguished on a behavioural level in such settings remains an open question [21].

Overall, our article, therefore, makes the following contributions to the existing literature. First, we propose a framework for analysing interaction patterns based upon time-stamped data collected with wearable sensors going beyond self-reported, retrospective perceptions of team behaviour. To do so, we provide a practical application of the Dynamic Actor Network Models (DyNAM) [34]. While DyNAM models have been used in the social sciences to explore the relation between social isolation and depressive symptoms [35], to understand debate networks in the European Parliament [36], or political discourse surrounding nuclear energy in Germany [37], it has not been applied to the analysis of interaction data in small teams. Our study is the first to apply a hierarchical DyNAM to real-world settings, analysing interaction patterns of 11 small Research & Development (R&D) teams across three organisational contexts: universities, research labs, and a private company. The highly granular, week-long behavioural data collected in each team enables us to show how interaction choices produce and reproduce interpersonal status hierarchies on the group-level, and how these vary across organisational contexts.

Secondly, our analysis contributes to the diversity literature by demonstrating the value of a configurational approach to team processes [38]. By demonstrating how specific dyadic configurations shape the frequency of social interactions, we provide evidence for the value of examining teams through the lens of unique socio-demographic pairings among team members. This configurational perspective further enables researchers to disentangle gender-based social categorisation effects from gender-based status effects during team interactions—two mechanisms that prior studies often conflate [27,29,31]. Building on this approach, we position our main contribution as offering a concrete example of how to investigate gender- and status-related dynamics at the behavioural level of dyadic team interaction.

In what follows, we first outline the current understanding of how gender and status effects condition team interaction behaviour. Second, we introduce the literature on Dynamic Actor Network Models, before outlining how both perspectives will be combined to analyse behavioural data collected with wearable sensors. Towards this goal, we formulate three hypotheses to test the effects of gender-based social categorisation and gender-based status effects on team interaction. The conceptual framework is then complemented by the methods section which includes a description of our data and the mathematical framework of the hierarchical Dynamic Actor Network Model. We then present the results before closing the article with a brief discussion and the limitations of this research.

## Theoretical background: Gender and status effects in teams

Most research on the benefits and drawbacks of team diversity is grounded in three main theoretical perspectives. The information-processing perspective [39] emphasises the value of diversity in teams, whereas the similarity-attraction perspective [40] highlights how in-group bias can increase coordination costs, ultimately diminishing the benefits of diversity. Finally, status characteristics theory examines how stereotype bias affects team dynamics, which also can undermine the benefits of diversity [29]. While the following paragraphs introduce briefly the core idea of each theoretical perspective, the reliance on wearable sensor data to track team interaction calls special attention to the similar-attraction theory and status characteristics theory, as only these frameworks generate empirically testable predictions concerning interaction behaviour among team members. In addition, while the diversity literature does address different socio-demographic and functional attributes—such as ethnicity, age, or education among others—this article focuses primarily on gender diversity, as gender remains one of the most salient and widely studied dimensions influencing team dynamics and organisational outcomes [41,42]. Overall, the conceptual framework clarifies how gender and status differences shape face-to-face team interaction, which in turn serves as antecedent to a range of team-level outcomes, including improving science [43], or generating innovative product solutions [44].

First, the information processing perspective on diversity in teams provides the overall rationale for the positive effects of work group diversity [39]. It is grounded in the value-in-diversity hypothesis [45], which proposes that categorical differences among team members—whether personal, social or functional—harbour a broader variety of task-relevant knowledge, skills, and experiences that ultimately lead to higher-quality solutions. By pooling the assets of many, teams gain in flexibility and innovation as they need to integrate a diversity of perspectives and ideas which also helps them to avoid premature consensus. Especially when faced with complex problems in high-stakes environments, a diverse set of skills and experiences amplifies the space of potential solutions available, thus increasing the chance of generating higher-quality and more innovative results [46].

While the information integration perspective has been widely used to argue for the benefits of gender diverse teams, for example in producing more novel and higher impact ideas [47,48], it does not include a theoretical model how diversity attributes such as gender condition team interaction [17,49]. While some research have shown that gender balanced teams perform better due to the higher levels of social perceptiveness and parity in conversational turn-taking [50], there is no solid conceptual link between gender and behavioural differences affecting information integration. Rather, the effects of gender diversity depend on other, moderating variables such as psychological safety [51], active inclusion [52], or skilled leaders [53] which are necessary for information integration. Hence, although the information-processing perspective provides a broad framework for understanding the benefits of (gender) diversity, it does not, in the context of this article and its reliance on behavioural interaction data, yield testable hypothesis. By contrast, the role of gender in shaping team interaction needs to be addressed through the similar-attraction paradigm on the one hand and status characteristic theory on the other.

Second, the social categorisation perspective views group dynamics as shaped by cognitive processes through which individuals define themselves and others based on salient social attributes such as gender or race [54–56]. These categorisations foster in-group identification and out-group differentiation. As Tajfel [57] and others in his wake have shown, the simple fact of belonging to a social group creates in-group favouritism and out-group bias and discrimination. Complementing this insight, similarity-attraction theory suggests that perceived similarity of oneself with others enhances interpersonal comfort and information exchange, while dissimilarity may disrupt it [40,58]. In short, people in general are attracted to and seek interaction with others they perceive as similar. The categorisation-elaboration model [46] integrates the effects of social categorisation with the information-processing perspective, by arguing that in-group bias can undermine the potential benefits of diversity.

Importantly, from a behavioural perspective on team interaction, social network scholars have gathered ample evidence regarding the validity of the similar-attraction paradigm in terms of assortative tie formation or homophily [59,60]. Gender homophily has been shown to drive tie formation among pupils in schools [61], online social networking sites [62,63], or collaboration and publication networks among researchers [64,65]. Concerning face-to-face interaction, the social psychology of in-group and out-group dynamics can be observed via quantifiable behavioural differences such as frequency and duration of interactions which give rise in turn to corresponding social ties and social network measures [24,66–68]. Recent studies using face-to-face contact diaries or sensor-based measures confirm this view, showing that gender homophily can be inferred from behavioural indicators such as spatial proximity, or number and duration of calls [69–72].

The third major approach to gender diversity in teams draws on social-psychological research on the role of (gender) stereotypes—a perspective largely overlooked by diversity scholars [29]. Gender stereotypes are culturally shared beliefs about traits or roles associated with men and women, shaping expectations and behaviours across many domains of social activity. In Western societies, gender stereotypes portray men as more agentic and competent, whereas women are seen as more communal and less competent [73,74]. Through these characteristic "Big Two" stereotypes (agency/competence vs. communion/warmth), gender provides a fundamental lens through which we process, perceive and navigate the social world [75]. Crucially, these gender stereotypes express a value hierarchy where (male) competence is in general higher valued than (female) warmth. As a result, (white) men enjoy in general higher status compared to women

or persons belonging to a minority group, given the widely shared cultural belief in higher task related competence of men [76–79]. Although women are no longer viewed as less intelligent than men [80], this historical shift in gender stereotypes "is not eroding the higher status currently associated with masculine traits and behaviors and the relative devaluation of stereotypically feminine traits and behaviors" [81]. This gender-based status hierarchy has consequences for team effectiveness, as it may collide with professional status cues—such as tenure, seniority, education, or contract type [82,83]—which serve as a critical coordination mechanism for information sharing and integration within teams.

Status is relevant for team interaction from a professional- as well as a gender-specific perspective. Status, in general, is defined as the prestige, esteem, and respect an individual enjoys in the eyes of others [84,85]. Status hierarchies within groups emerge based upon the expectations that actors form regarding the task related competence of its members [86,87]. Perceived professional status differences in work groups thereby serve as a tacit coordination mechanism, indicating who among its members is the most competent person for a job [30,88]. Ideally, the status hierarchy within groups reflects the real expertise and contribution of team members to the group. Thus, individuals primarily orient their interactions toward high-status team members [89,90], who typically occupy central network positions reflecting their contributions and group importance. Advice seeking, for example, targets higher-status actors [27,91,92], and ties to such alters often confer additional status, reinforcing their network centrality [93]. Overall, high-status members—such as leaders or senior staff—are expected to interact more frequently than low-status actors like newcomers or juniors [94,95].

However, as argued, gender as a diffuse status cue potentially undermines such an ideal scenario [96]. For example, incongruencies between the perceived status-level based on professional rank compared to the status-level based on gender can give rise to a "gender-status mismatch" which occurs when women in senior positions clash with widely held cultural beliefs about men's superior competence [97,98]. The gender-status mismatch then can undermine the benefits of diversity as when incorrect attributions of competence harm information processing and integration [99]. Crucially, when considering the behavioural implications of gender-based diffuse status cues on interaction, status characteristic theory makes clear predictions regarding gendered interaction patterns: men, as members of a high-status group, typically dominate gaze and gestures, speak more frequently, interrupt more often, claim credit more readily compared to women [100–103]. High gender status is mirrored in both verbal and nonverbal behaviours, granting high-status actors more opportunities to act, greater task contributions, and more positive evaluations of those contributions [104]. As a result, we expect that gender-based status expectations increase the likelihood to orient one's interaction towards the higher status group, namely men. Women within a specific rank are more likely to interact with men (higher gender status group) and are less likely to interact with other women (lower gender status group). In summary, professional-based status should increase the likelihood of interaction for women and men alike, while gender-based diffuse status cues should reduce the likelihood of interaction for women but increase them for men.

## Analytical approach: Relational Event Models to study social interaction among team members

As outlined in the introduction, the inconsistent findings of prior team diversity research are partially attributable to the dominance of static collectivist accounts that assume simple, linear relationships between team-level diversity predictors and team-level outcomes [5]. Shifting towards a microdynamic approach foregrounds instead an interaction-centric, temporal explicit account of how relational (dyadic) configurations among team members produce higher-level, emergent outcomes. According to van Dijk et al. [21], the core thrust of a microdynamic perspective is to understand how team-level phenomena emerge from the ongoing organising processes among its members, where outcomes such as team performance are shaped by individual attitudes, behaviours, and interactions [105]. At its core, a microdynamic view situates interaction as the behavioural foundation of teamwork, from which all shared cognitive structures must emerge [14].

A logical implication of the primacy of interaction among team members is the adoption of a configurational perspective: because individual socio-demographic attributes (e.g., gender, age, race) shape how members perceive and respond to one another [106], it is ultimately the pattern of dyadic configurations across team-member pairs that drives team-level

outcomes. Being able to better distinguish how different team member pairs perceive and react to each other enables one to examine more closely how gender and status effects play out in each team. As argued by van Dijk et al. [21], the unique contribution of the microdynamic perspective consists of introducing a "perceiver–target mechanism (the MIDST model)" into the diversity literature. Such a "perceiver-target" model enables one to distinguish for example homophily effects (based on the similar attraction paradigm) from stereotype-based attributions of competence or warmth, which trigger reinforcing loops of behaviour and performance. Disentangling, however, homophily from gender-based status effects without losing sight of their temporal dynamics requires more powerful network analytical approaches, to be introduced in the following section.

Relational Event Models (REM) [34,107] enable researchers to build and test these increasingly sophisticated models of interaction sequences (for reviews see [108,109]). Unlike earlier network research that condenses multiple social contacts into relatively coarse snapshots of binary ties, REMs preserve the dynamic richness of social interactions by tracking who is interacting with whom and when [110,111]. In order to do so, a REM conceptualises social interaction as a time-stamped sequence of discrete sender→receiver events (e.g., a phone call between A and B), where the likelihood of a given event depends on both the endogenous history of the network (e.g., previous calls) and external features to the event history itself (e.g., organisational context) [107,112]. Common endogenous covariates—such as reciprocity, inertia, recency, or repetition—account for the micro-temporal sequencing of events where past events influence what happens next. Exogenous covariates in contrast account for features external to the relational event history itself such as differences in organisational contexts (e.g., laboratory vs. private offices), dyadic attributes (e.g., similar team roles), or socio-demographic attributes (e.g., age, nationality, gender). By integrating exogenous variables at global, node, or dyadic levels, relational event models can assess how interaction patterns vary across configurations of interpersonal attributes. Overall, REMs are particularly suited to advancing a microdynamic perspective in diversity research as they enable one to move beyond assumptions of temporal homogeneity and aggregated team measures to provide a precise analysis of configurational and temporal aspects of social interaction [109,113].

Given the benefits of modelling both the temporal unfolding of social interaction and its contexts, relational event models have gained increasing popularity [114,115]. Exemplary studies in the social sciences have applied REM to understand the radio communication patterns in response to the 9/11 attacks on the World Trade Centre [116], critical team coordination [114,117], congressional collaborations [118], or exchanges in financial markets [119,120] to name just a few. Scholars thereby use different data sources such as digital traces of Wikipedia edits [121], face-to-face interaction among freshman students [112], or interhospital patient transfers [122]. What qualifies these data sources as relational event sequences is the fact that edges are continuously activated and deactivated, and that no single event on its own is characteristic of the network state [95]. With the exception of Hoffman et al. [123], to the best of our knowledge no other study has applied so far a relational event model to analyse social interactions using data collected from wearable sensors.

Among the REM models, the Dynamic Network Actor Model (DyNAM) [34] investigates relational events as an actor-oriented process in which actors choose interaction partners based on their preferences and the available opportunities. For the present analysis, we employ DyNAMs for coordination ties, which are specifically designed to analyse undirected relational events where both actors must agree to the interaction [124]. This formulation is particularly appropriate for face-to-face interactions captured through wearable sensors, where co-presence and mutual engagement are required for an interaction to occur.

The coordination model conceptualises undirected social interactions as a three-step decision process. First, an actor $i$ considers proposing an interaction. Second, $i$ selects a potential partner $j$ from the available actors. Third, the proposed partner $j$ evaluates whether to accept the interaction with actor $i$. A tie is observed only when both the proposal and acceptance occur. Although DyNAM models these coordination ties as a three-step process, the sensor data used in our study does not distinguish between the interaction proposal by the actor $i$ and the interaction acceptance by the actor $j$. Infrared face-to-face interactions indicate co-presence among actors. Infrared detections resemble, in this sense, a "dictatorial"

[125] means of two-sided tie formation, where interaction partners impose ties on each other without explicit agreement. To distinguish an aleatory passing-by among team members without an actual interaction taking place, we require at least 2 detections within a time window of 75 seconds.

The probability that actor $i$ proposes an interaction to actor $j$ is modelled using a multinomial logit specification:

$$p(i \rightarrow j;\ y,\ \beta) = \frac{exp(\beta^T s(i, j, y))}{\sum_{k \in \mathcal{A}\backslash\{i\}} exp(\beta^T s(i, k, y))}$$

where $y$ is the process state, encompassing all relevant information from the event history up to time $t$. $\mathcal{A}$ is the set of actors available to receive an event at time $t$. The vector $s(i, j, y)$ contains statistics characterising the dyad $\{i, j\}$ and its embedding in the network structure. The statistics capture potential mechanisms driving interaction choices such as inertia (tendency to repeat past interactions), degree (tendency to interact with popular actors), and homophily (tendency to interact with actors sharing attributes).

The parameters β quantify the relative importance of different network mechanisms in event creation. Positive coefficients indicate that the corresponding statistic increases the probability of proposing or accepting an interaction, while negative coefficients decrease this probability.

To simplify estimation, the coordination model assumes a constant rate of event proposals (first step). Therefore, the model focuses on explaining which pairs of actors engage in a relational event rather than when events occur. Under this assumption, the probability that the next observed event is from the pair $\{i, j\}$ rather than any other pair of actors is:

$$P_{i \leftrightarrow j}(y) = \frac{p(i \rightarrow j;\ y,\ \beta)p(j \rightarrow i;\ y,\ \beta)}{\sum_{k,l \in \mathcal{A};\ k < l} p(k \rightarrow l;\ y,\ \beta)p(l \rightarrow k;\ y,\ \beta)}$$

When analysing interaction sequences across multiple contexts (e.g., different offices, teams, or departments), actors and dyads may display heterogeneous behavioural patterns that are not fully explained by observed covariates. Random effects provide a framework for modelling such unobserved heterogeneity while sharing statistical strength across contexts [126].

In the random effects extension, the linear combination used in the multinomial logit specification is decomposed into fixed and random effect components:

$$\beta^T s_h(i, j, y) + \alpha_g{}^T s_m(i, j, y)$$

where the parameters $\alpha_g$ are the random effect coefficients for team $g$ related to the vector of statistics $s_m$. The random effects are assumed to follow a multivariate normal distribution: $\alpha_g \sim N(\gamma, \Sigma)$, where $\gamma$ is the average relative importance of the statistics across all the teams, and $\Sigma$ is the covariance matrix capturing both the variability of each effect across teams and the correlations between different effects. This specification allows network mechanisms to vary systematically across organisational contexts while borrowing statistical strength from the entire sample. A more general description of the random effects model specification is described in [126].

## Hypothesis

Combining our theoretical framework on the role of gender and status in teams with a relational event model we formulate the following set of hypotheses.

First, based upon the similar-attraction paradigm described earlier, we assume that gender homophily increases the likelihood of team interaction. From a behavioural perspective this implies that the preference for similar others can be observed via measurable differences in frequency and duration of interactions. Across our 11 R&D teams, we suppose that same gender dyads are more likely to interact than mix-gender dyads, expressed as our first hypothesis:

*Hypothesis 1: Team member dyads with the same gender are more likely to interact than mixed gender dyads*

Exploring gender homophily with a hierarchical DyNAM model implies to test H1 through a set of two sub-hypothesis, which operationalise gender homophily at the dyadic level:

*Hypothesis 1a: Same-gender women dyads are more likely to interact than mixed gender dyads*

*Hypothesis 1b: Same-gender men dyads are more likely to interact than mixed gender dyads.*

The second hypothesis concerns the role of professional status on team interaction. As described, status hierarchies in groups emerge from members' expectations about each other's task-related competence. Given the link between professional seniority and status [82], we propose that senior team members—who occupy high-status roles—are more central in the group's interaction network, engaging more frequently with others as the preferred access point for team-related matters:

*Hypothesis 2: Being a senior team member (high-status) increases the likelihood of interaction compared to a junior team member (low status)*

Exploring the impact of seniority/ high status on team interaction with DyNAM requires the conversion of H2 into two sub-hypotheses on the dyadic level:

*Hypothesis 2a: junior team members (low status) are more likely to interact with senior (high-status) team members*

*Hypothesis 2b: Senior team members (high-status) are more likely to interact with other (high-status) senior members*

Finally, we formulate a third hypothesis to test the effect of gender-based status cues on interaction frequency. We stipulate that among team members that pertain to the same seniority level (either junior or senior), interaction frequency increases with others that pertain to the higher gender-based status group. We assume that women (lower gender-based status group) are more likely to interact with men (higher gender-based status group) within their seniority level, while men should tend to select other men (same higher gender-based status). Note that the difference between gender-homophily and status-based effects can only be observed for women dyads. Within the same seniority level, gender homophily exists to the degree women interact with other women, while gender-based status cues would exist to the degree that mixed-gender interactions (women as low-status group with men as high-status group) increase the odds of interaction. Men interacting with other men within the same seniority level do not enable us to distinguish between similar-attraction (men-men), or gender-based status (men-men). Given that we cannot distinguish gender homophily from status-based effects for men, we formulate the following hypothesis concerning gender-based status effects for women:

*Hypothesis 3: Within a given seniority level, women prefer to interact with team members of the higher gender-based status category, namely men.*

We operationalise this hypothesis via one sub-hypotheses for junior and senior women who would be more likely to interact with other junior/ senior men:

*Hypothesis 3a: Among junior team members, junior women prefer to interact with junior men*

*Hypothesis 3b: Among senior team members, senior women prefer to interact with senior men.*

In what follows we describe our empirical setting and sample data, measures, and model specification to test these three hypotheses.

## Methods

### Empirical setting & data collection

Data was collected across 11 small R&D teams using wearable sensors. As show in Table 1, participating teams belonged to different organisational settings: three teams operated within the private sector, belonging to the same company; five teams operated in university associated research labs, while three additional teams were in academic university settings.

As Table 1 indicates, the sample of research teams is quite heterogeneous in terms of organisational settings, country and disciplinary background. Considering each of these, research has shown that national culture has a relatively small influence on organisational culture [127]. In addition, country-level gender equality indicators between the UK and Spain

**Table 1. Overview of R&D team characteristics, organisational settings, discipline and country context.**

| Team ID | Discipline | Organisation | Country | Total members |
|---|---|---|---|---|
| 1 | Biomedical Engineering | University | Spain | 8 |
| 2 | Biomedical Engineering | Research lab | Spain | 10 |
| 3 | Biomedical Engineering | University | Spain | 8 |
| 4 | Biomedical Engineering | Research lab | Spain | 9 |
| 5 | Biomedical Engineering | Research lab | Spain | 11 |
| 6 | Energy Engineering | University | UK | 10 |
| 8 | Transport Engineering | Private company | UK | 8 |
| 9 | Materials research | Research lab | Spain | 17 |
| 10 | Materials research | Research lab | Spain | 8 |
| 7A | Transport Engineering | Private company | UK | 7 |
| 7B | Transport Engineering | Private company | UK | 7 |

show small differences regarding gender gaps in education, political empowerment, health, and economic participation [128]. This also reflects similarities in terms of gender ideology across these countries, as both the UK and Spain score highest on the "egalitarian" and "choice egalitarian" regime [129]. We therefore expect minimal influence of country-level differences regarding gender on our behavioural measures. Much stronger effects are likely to arise from the organisational setting, which shapes face-to-face communication patterns directly through factors such as office and laboratory layouts or the degree of co-dependence among team members (e.g., teaching vs. collaboration in labs). For instance, open-plan office spaces facilitate more frequent and shorter face-to face interactions, whereas interactions are less frequent in traditional environments with private offices [130,131]. We account for these differences by examining how our effect parameters vary across the three organisational settings (see *Measures* for further details).

Disciplinary differences are also relevant from a gender perspective, as some fields are highly male- or female-dominated. Token status for women (or men) exists where the percentage of women drops below 15% [132]. However, the R&D teams in our study were deliberately selected from disciplines where mixed gender teams are the norm. For example, biomedical engineering combines male-dominated engineering with medical expertise where many women are found. Similar, transport research combines male-dominated (automotive) engineering with women's higher interest in the environment and sustainable transportation. These combinations create relatively gender-balanced work environments in the participating teams, and we therefore do not expect disciplinary background to substantially affect team interaction.

Furthermore, Team 7 requires special attention, as we split this team into two separate entities (Team 7A and Team 7B). Members of Team 7 belong to the same research group but are based in different cities, making physical contact impossible without traveling. Since we record face-to-face interactions using wearable sensors and do not consider other means of communication such as emails or online conferences, team 7A and 7B function effectively as two distinct work groups.

Teams were recruited following the same standard procedure. A face-to-face briefing session was held with each R&D team before the start of the field period in order to explain the objectives of the study, data to be collected, the methods, foreseen results, and how to handle the Sociometric badges (turning on/off, how to wear the badge, pick-up/drop off). Research statements and consent forms were emailed to all team members before the briefing, noting the option of dummy badges for those who preferred not to participate without being singled out. Written informed consent was obtained from each participant prior to data collection, and no dummy badges were requested.

Interaction data was collected over five consecutive working days using the infrared sensors of Sociometric badges (Humanyze, Boston, USA). The badges, worn on a neck strap, are slightly larger than a standard conference badge

(9.5×6×1.3 cm). Although equipped with multiple sensors—including Bluetooth, microphone, and accelerometer—only infrared data was used for this analysis. An infrared sensor of Sociometric badge A detects another device B if it is within the range of 1–1.5 meters and oriented towards the other within a 30º-degree cone [133]. Previous validation studies suggest the distance of infrared detections with Sociometric badges works well up to 2 meters but not beyond [18]. Because infrared sensors can register up to 50 detections per minute between two badges at distances under 2 meters, they are well suited for capturing workplace interactions. However, raw co-presence counts are often driven by scan frequency rather than distinct social exchanges. For example, 50 detections within a minute between two team members represent a single continuous interaction, not 50 separate events. We follow the recommendations of Elmer et al. [134] who found the best fit (88,9% of accuracy) between infrared detections and video coded social interactions at a maximum aggregation threshold of 75 seconds. Successive detections within the same 75 second time windows have thus been aggregated to a single interaction.

In addition to the interaction data, a short questionnaire was also sent out to all team members during the field-period to collect socio-demographic information, including age, gender, highest qualification, role in the team, and duration of team membership. Each team member was also asked to indicate their friendship- and advice-seeking ties with each of their colleagues.

The research was approved by the ethics committee of the Universitat Oberta de Catalunya and passed the Ethics Review as part of the grant agreement signature process with the European Commission.

## Measures

We first introduce the variables and measures to test our main hypothesis, followed by additional variables to account for alternative explanations of the observed social interaction patterns. Our DyNAM model includes both exogenous and endogenous covariates, each briefly described in what follows. An overview of descriptive statistics of socio-demographic variables across teams is available in supplementary file S1–Table 1; basic face-to-face interaction counts and network visualisations for each team is also available in supplementary S1 File.

*Gender.* The gender of team members was measured via a binary variable (*Woman*, *Man*). An *Other* option was available but not used by any respondent. All teams are fairly gender balanced, leading to an overall distribution of 45% women and 55% men (see Table 1 in S1 File). Gender is converted into two exogenous covariates that measure the homophily of $\beta_4 = man\text{-}man$, and $\beta_5 = woman\text{-}woman$ dyads. In our final model, a positive beta-parameter for $\beta_5 = woman\text{-}woman$ would indicate that same-gender dyads of women are more likely to interact than mixed-gender dyads, i.e., woman-man.

*Status.* We measure the professional status of a R&D team member through the role they occupy [32], classifying them into *Senior* (including *Team leader*, *Senior researcher*, and *Postdoc*) and *Junior* roles (*Research or lab assistant/ technician*, *PhD Student*, *MA Student*, *Administrative assistant*). These team roles are indicative of a person's expertise, and hence likely status. Seniority roles are incorporated as exogenous, dyadic covariates into our model and allow us to estimate parameters $\beta_9 = junior\text{-}junior$, and $\beta_{10} = senior\text{-}senior$ dyad. A positive beta-parameter $\beta_9$ in our final model indicates that junior-junior dyads are more likely to interact than mixed-seniority dyads.

In addition to the main predictors gender and seniority the hierarchical DyNAM incorporates additional measures to account for alternative explanations of interaction patterns.

*Team tenure* is closely related to seniority and hence a potential cofounder for the status of team members. The longer an individual forms part of the team, the more experience and expertise she or he accumulates, and the higher their status. We thus incorporate team tenure as an additional exogenous covariate, as it not necessarily overlaps with seniority. Team tenure is measured for each team member in months elapsed since joining the team. Three discrete tenure categories are constructed to ease interpretation, consisting of relative *Newcomers* (tenure ≤ 12 months), *Consolidated* members (12 < tenure ≤ 72 months), and *Long-term*, founding members (tenure > 72 months). These tenure categories are incorporated as $\beta_{20} = same\ tenure$, where a positive parameter in our final model indicates that same tenure dyads are more likely to interact than mixed-tenure dyads.

*Age* of team members. Although available as continuous variable in the original data set, we use a discrete variable for our models to ease interpretation. We construct three age bands: *Young* (<= 30 years), *Middle* (31–45 years) and *Elderly* (> 46 years). According to Eurostat [135], the mean age of women at childbirth in 2018 was 30.8 years. Thus, our age bands group team members according to family formation events (no family formation before 30 years of age, family formation between 31–45 years of age), assuming that family formation marks an important life course event that shapes interest, concerns, and availability of team members during interaction. The mean age varies across teams, with some teams having relatively younger members (e.g., Team 5, mean age = 29 years) and others having relatively older members (e.g., Team 1, mean age = 46 years). Age is incorporate as exogenous, dyadic covariate $\beta_{19}$ = same-age category. A positive value of this beta-parameter indicates that dyads within the same age group are more likely to interact than dyads of mixed-age categories.

*Advice seeking.* Participants indicated the frequency with which they ask their colleagues for 'work related advice' (1 = *Never*, 2 = *Rarely*, 3 = *Sometimes*, 4 = *Very often*, 5 = *Always*). A dichotomous, directed matrix is constructed, indicating an advice seeking tie between A and B in case the rating of A is above 3 (and vice versa, for B rating A). We explore how the presence of $\beta_7$ = advice-seeking ties increases or decreases the likelihood of interaction among team members.

*Friendship.* Participants were asked to indicate the frequency with which they spend time socially with their colleagues outside the lab/office (1 = *Never*, 2 = *A few times a year*, 3 = *A few times a month*, 4 = *A few times a week*, 5 = *Daily*). A dichotomous, directed matrix is constructed, indicating a friendship tie nomination between A and B in case the rating reported by A is above 3 (and vice versa, for B rating A). We explore how $\beta_6$ = friendship ties increase the frequency of interaction among team members.

*Co-location.* Additional information was collected by researchers about the co-location of team members. If team members work in the same laboratory or office, the collocation matrix of the corresponding dyad is set to 1 and to 0 otherwise. Team members that share office space are more likely to interact [60]. In addition, the set of available interaction partners is also adjusted considering the actual presence or absence of team members in the office. We account for the absence or presence of team members at their workplace and hence the opportunity to interact (or not) based upon the duration of their Sociometric badges being switched on. Parameter $\beta_8$ = co-location estimates how shared office or laboratory spaces increments the likelihood of interaction among team members.

*Organisational context*. Many aspects of organisational context have been shown to affect teams, for example in terms of research performance [136,137], the impact of diversity [138] or learning [139]. In this study, where social interaction data was collected via wearable sensors, organisational context is critical. Physical layout (e.g., fixed desks vs. open-plan laboratories) and team routines (e.g., teaching vs. collaborative lab work) strongly shape interaction patterns. Among the 11 R&D teams, contact frequency is highest in the private company with open-plan office space, lower in research labs, and lowest in universities, where individual offices and teaching duties limit co-presence [20]. Given the importance of the organisational context for interaction frequency [130,140,141], we distinguish explicitly between university, research lab, and private company when estimating parameters with DyNAM. This enables us to test if our three hypotheses hold up across organisational contexts.

In addition to these exogenous covariates, the DyNAM model incorporates three endogenous covariates to account for the temporal effects of face-to-face interaction, namely inertia, degree, and recency.

*Inertia* assesses whether the occurrence of past interactions increases the likelihood of future interactions. In a knowledge-intensive work environment, such as R&D teams, inertia can reflect ongoing collaborations among team members, for example, advice-giving relationships between postdocs and PhD students. Given the continuity of such a supervisory relationships, team members who have interacted more frequently in the past are also more likely to interact in the future. A positive parameter $\gamma_1$ = inertia in our final model indicates that team members that have interacted in the past are more likely to do so again in the future.

*Indegree* in the network of past event serves as an indicator of an actor's centrality, assessing whether central actors are likely to be part of future events. The degrees are normalised by the size of the team. A positive parameter $\beta_2 = degree$ indicates an increased likelihood of actors to interact with central actors in the network of past events.

*Recency* addresses the commonly observed pattern in empirical applications where recent events wield more influence on the occurrence of subsequent events than those occurring further in the past. A positive parameter $\beta_3 = recency$ indicates that two consecutive interactions are more likely to happen between the same dyad rather than different ones.

Several of the survey-based, socio-demographic variables such as age, team tenure, friendship and advice seeking ratings have missing data. To impute missing values for these variables, the Multiple Imputation method [142] is used. A hierarchical regression model with random intercepts varying by team is used to consider the inherent multilevel structure of the data. Gender and a binary indicator of seniority roles are used as auxiliary variables that yield a distribution of plausible values to impute missing scores. For age and team tenure, we assume their logarithms follow a multivariate normal distribution, following McElreath's [143] treatment of missing values for continuous covariates. For dyadic variables like advice-seeking and friendship, we employ the hierarchical p2 model [144]. This model is particularly suited for dealing with the complexities of social network data, including the interdependence of relationships and the incorporation of individual and dyadic characteristics. The imputation model for these variables considers both dyadic covariates—such as same gender and seniority level—and individual characteristics, including the gender and senior role of both the sender and receiver of the tie.

## Model specification

The linear predictor for the coordination DyNAM with random effects is as follows:

$$\alpha_{1g}^c s_{\text{inertia}}(i, j, y) + \sum_{k=2}^{K} \beta_k^c s_k(i, j, y)$$

This formulation allows the random effects and fixed effects parameters to vary by the organisational context represented by the index *c*, which takes the values *uni* (University), *lab* (Research Lab), or *bus* (Private Company).

Preprocessing of the event data uses the goldfish R package [145], which enables the computation of effects used as explanatory variables in the model. The analysis follows a Bayesian inference framework [143] and employs the Hamiltonian Monte Carlo (HMC) algorithm as implemented in Stan [146], applying weakly informative priors similar to those recommended by [126]. Convergence of the HMC algorithm was assessed through trace plots as a diagnostic of mixing chains and statistics, including the rank-normalised split-*R* statistic and effective sample size, as outlined by [147]. For each imputed dataset, we drew 500 samples from the posterior distribution of the effect coefficients using two parallel chains and then computed the posterior summary statistics from the pooled distribution across all imputed datasets. In alignment with the recommendations proposed by [148], we generated 100 imputed datasets to produce these posterior summary statistics.

To interpret the fitted models, we summarise the posterior distributions using the median as a point estimate and the 95% equal-tailed credible interval (CI), which contains the central 95% of the posterior mass and excludes the lowest and highest 2.5%. We also report the probability of direction (pd), which quantifies directional certainty by computing the proportion of posterior samples that share the same sign as the median [149].

Hypotheses are evaluated by jointly considering the probability of direction and the credible intervals. We classify evidence strength as follows: strong directional evidence when the 95% CI excludes the reference value (zero for β coefficients, one for probability ratios), this necessarily implies pd ≥ 0.975; moderate directional evidence when pd ≥ 0.95; weak directional evidence when pd ≥ 0.90; and insufficient directional evidence when pd < 0.90. Together, these metrics indicate whether an effect is likely to exist and provide its plausible range, allowing us to assess the strength of evidence for each hypothesis.

To assess organisational heterogeneity, we report posterior statistics for the largest pairwise difference in β coefficients between contexts (Δβ), along with 95% CIs and probability of direction. These direct comparisons quantify whether observed cross-context differences are substantive and provide their plausible range.

Furthermore, interaction coefficients cannot be interpreted in isolation from their corresponding main effects. To assess how status modifies gender homophily, we calculate probability ratios (PR) that compare the likelihood of interaction between two prototypical dyads (e.g., same-gender versus mixed-gender) that differ only in the attribute of interest. This approach enables us to quantify how differences in actor and relational attributes influence the probability that a dyad interacts next. Conceptually, this procedure parallels the use of probability ratios in Stochastic Actor-Oriented Models (SAOMs), where they are used to illustrate how specific changes in covariates drive tie formation [150]. The PR is calculated as the ratio of the exponentiated linear predictors for two hypothetical dyads. For the DyNAM coordination, the linear predictor is scaled by a factor of two to account for the model's mutual agreement mechanism, in which the effect contributes to both $i$'s proposal to $j$ and $j$'s proposal to $i$. In addition, it is important to note that this calculation relies on the numerator of the multinomial logit specification. Consequently, interpreting the PR as a direct comparison of probabilities implies a *ceteris paribus* assumption: it presumes that the opportunity structure (the denominator representing the aggregate attractiveness of alternative partners) is roughly equivalent for the actors involved. Thus, the PR isolates the structural preference for specific dyadic configurations, serving as a proxy for relative probability under the assumption of comparable choice sets.

A prototypical dyad is defined by starting from the model's baseline dyad—one that has mixed seniority status and no advice or friendship ties—and then modifying it according to attributes and the relationship focus relevant to the interpretation. For example, a women-only/ mixed-gender probability ratio compares women-women to mixed-gender dyads under the condition that neither have advice or friendship ties and both have mixed seniority. Meanwhile, a women-only/ mixed-gender juniors probability ratio compares women-women with mixed-gender dyad when both actors occupy junior positions and do not have advice or friendship ties.

## Results

Posterior summaries of DyNAM β-parameters are reported in supplementary S2 File–Table 1; probability ratios in S2–Tables 3–4; and random effects by team and their standard deviations in S2–Tables 7–8, respectively.

Fig 1 displays half-eye plots of the posterior distribution for the PRs of gender homophily by seniority level. The upper half shows the posterior densities; the lower half displays summary statistics: the median (dot), the 50% equal-tailed credible interval (thick line), and the 95% equal-tailed credible interval (thin line). This visualisation allows inspection of both the full posterior and conventional interval-based summaries.

Examining the effects of *gender homophily* under the similar-attraction paradigm of team interaction, we find strong negative effects in universities and research labs, but not in private companies. In universities, women-women dyads are half as likely to interact next as mixed-gender dyads ($\beta_5^{uni} = -0.35$, 95% CI [–0.66, –0.06], pd = 0.99; $PR^{uni} = 0.50$, 95% CI [0.27, 0.89]). The effect is stronger in research labs, where women-women dyads are approximately one-third as likely to interact next ($\beta_5^{lab} = -0.54$, 95% CI [–0.75, –0.33], pd = 1.00; $PR^{lab} = 0.34$, 95% CI [0.22, 0.51]). Private companies show insufficient evidence of any effect ($\beta_5^{bus} = 0.17$, 95% CI [–1.18, 1.21], pd = 0.61; $PR^{bus} = 1.40$, 95% CI [0.09, 11.20]). The organisational contrast between private companies and universities shows insufficient evidence of a substantial difference ($\Delta\beta_5^{bus-uni} = 0.52$, 95% CI [−0.86, 1.61], pd = 0.79).

For men-only dyads ($\beta_4$), we find inconsistent patterns across organisational contexts. Universities show weak evidence for a positive effect ($\beta_4^{uni} = 0.25$, 95% CI [–0.12, 0.55], pd = 0.90; $PR^{uni} = 1.66$, 95% CI [0.79, 3.02])—while the posterior median suggests men-men dyads are 1.66 times as likely to interact next, the wide credible interval reflects substantial uncertainty. Research labs show moderate evidence for a negative effect ($\beta_4^{lab} = −0.21$, 95% CI [−0.45, 0.03], pd = 0.96; $PR^{lab} = 0.66$, 95% CI [0.41, 1.06]), with men-men dyads approximately two-thirds as likely to interact next.

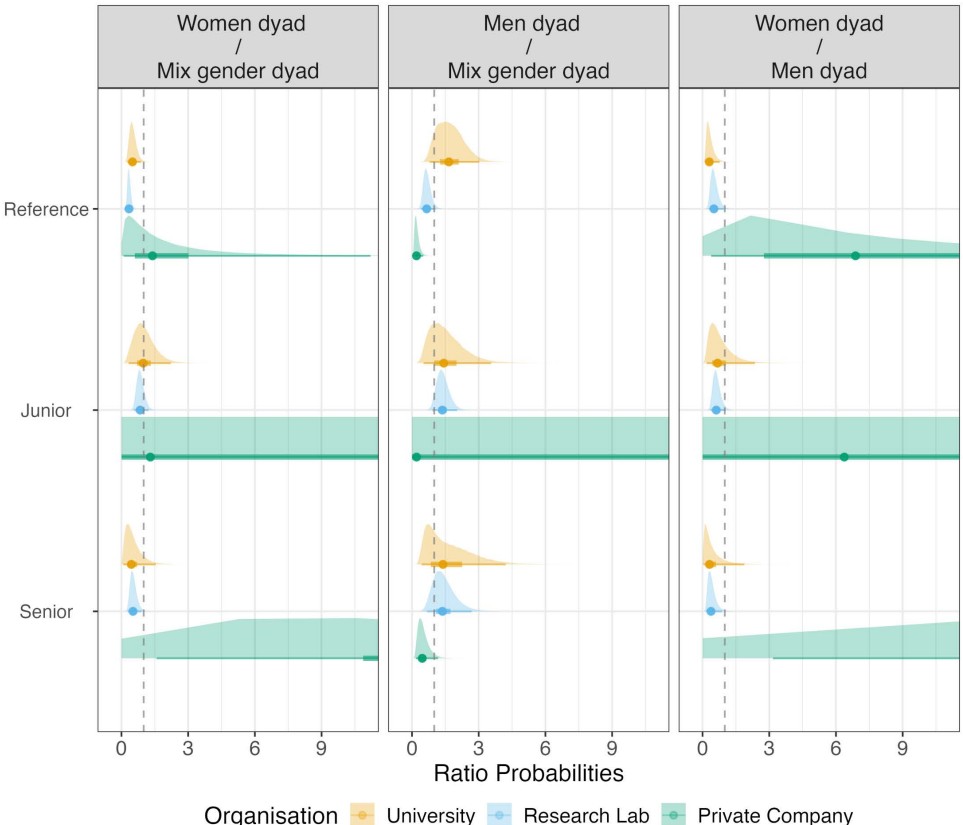

**Fig 1. Probability Ratio (PR) for gender-homophily by seniority composition.** PR > 1 indicates a higher probability for the dyad in the numerator to create an event next. Reference show baseline probability ratios without conditioning on seniority. Posterior distribution has been truncated for Private company. For numerical values see supplementary S2 File – Table 4.

Private companies provide the strongest evidence for a negative effect, with men-men dyads being one-fifth as likely to interact next as mixed-gender dyads ($\beta_4^{bus} = -0.80$, 95% CI [−1.28, −0.34], pd = 1.00; $PR^{bus} = 0.20$, 95% CI [0.08, 0.50]). The organisational contrast between private companies and universities shows strong evidence of a substantial difference ($\Delta\beta_4^{bus-uni} = -1.05$, 95% CI [−1.63, −0.45], pd = 1.00).

Overall, at baseline—absent advice or friendship ties and among mixed-seniority dyads—we find no evidence supporting H1 on gender homophily. Instead, the data reveal either no effect or negative homophily (reduced same-gender interaction), with substantial heterogeneity across both gender and organisational context. The similar-attraction paradigm does not hold for these baseline face-to-face interactions.

Moving on to our second hypothesis, we assess the role of professional status on team interaction. We test H2 regarding the increased likelihood of interaction among senior team members compared to juniors via two sub-hypotheses. First, H2a proposes that junior team members interact more frequently with senior members than with other juniors. As show in Fig 2, we find weak to moderate evidence supporting this pattern in academic contexts, and no evidence in private companies. In the university context, junior-junior dyads are 0.68 times less likely to interact compared to junior-senior dyads ($\beta_9^{uni} = -0.19$, 95% CI [−0.45, 0.07], pd = 0.93; $PR^{uni} = 0.68$, 95% CI [0.41, 1.15]), though the credible interval includes 1. Research labs show a similar but slightly weaker pattern ($\beta_9^{lab} = -0.13$, 95% CI [−0.27, 0.02], pd = 0.96; $PR^{lab} = 0.78$, 95% CI [0.58, 1.04]). Private companies show no clear pattern with substantial uncertainty ($\beta_9^{bus} = 0.41$, 95% CI [−0.28, 1.06],

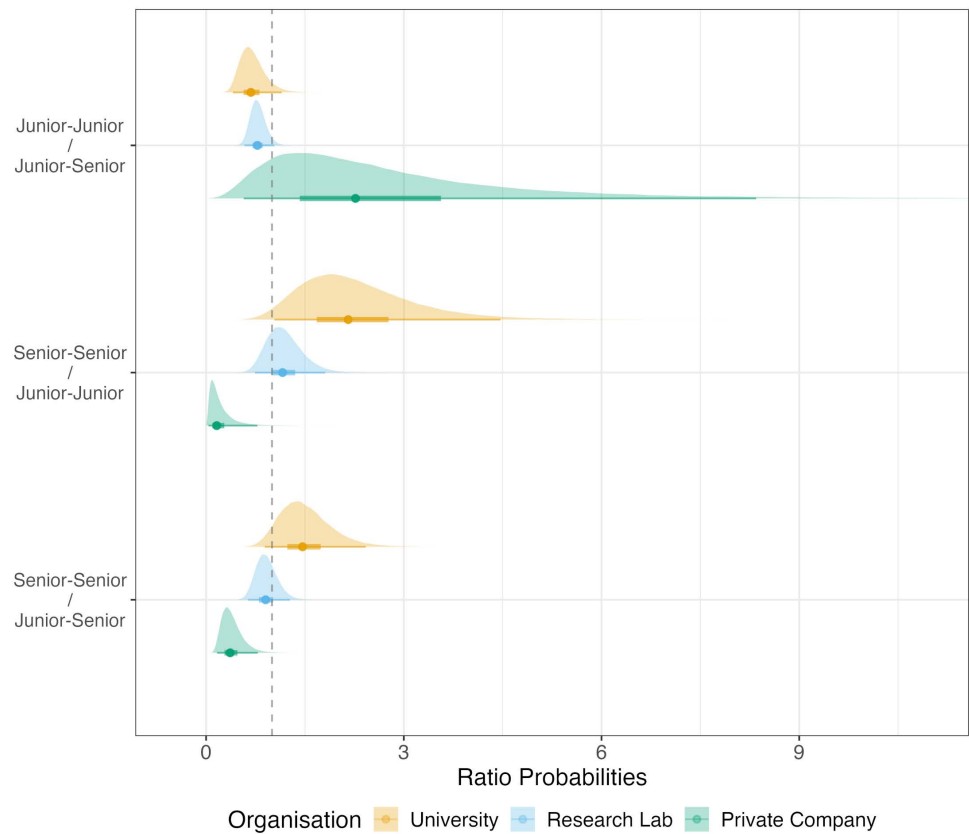

**Fig 2. Probability Ratio (PR) of professional status effects based upon seniority roles.** PR > 1 indicates a higher probability for the dyad in the numerator to create an event next. For numerical values see supplementary file S2 – Table 4.

pd = 0.88; $PR^{bus}$ = 2.27, 95% CI [0.57, 8.35]). The organisational contrast between private companies and universities shows weak evidence of a substantial difference ($\Delta\beta_4^{bus-uni}$ = 0.60, 95% CI [−0.14, 1.30], pd = 0.95). Overall, evidence for H2a is weak and context dependent.

Second, H2b proposes that senior-senior dyads interact more frequently compared to mixed-seniority dyads. As Fig 2 indicates, we find weak evidence in universities, insufficient evidence in research labs, and strong contrary evidence in private companies. Universities show that senior-senior dyads are 1.46 times as likely to interact next as mixed-seniority dyads ($\beta_{10}^{uni}$ = 0.19, 95% CI [−0.06, 0.44], pd = 0.93; $PR^{uni}$ = 1.46, 95% CI [0.89, 2.42]), though the credible interval includes 1. Research labs show no clear pattern ($\beta_{10}^{lab}$ = −0.05, 95% CI [−0.23, 0.12], pd = 0.72; $PR^{lab}$ = 0.90, 95% CI [0.64, 1.27]). Private companies provide strong evidence for a negative effect, with senior-senior dyads approximately one-third as likely to interact next ($\beta_{10}^{bus}$ = −0.50, 95% CI [−0.90, −0.12], pd = 1.00; $PR^{bus}$ = 0.37, 95% CI [0.17, 0.78]). The organisational contrast between private companies and universities shows strong evidence of a substantial difference ($\Delta\beta_{10}^{bus-uni}$ = −0.69, 95% CI [−1.16, −0.24], pd = 1.00). Overall, H2b receives weak and inconsistent support across organisational contexts.

Overall, evidence for H2—that senior team members interact more frequently with one another—varies substantially across organisational contexts. Universities provide weak support: junior-junior dyads are 0.68 times as likely to interact as junior-senior dyads, while senior-senior dyads are 1.46 times as likely (though both credible intervals include 1). Research labs show no consistent pattern for either comparison. Private companies show strong evidence against

senior-senior interaction (PR = 0.37), with uncertain patterns for junior-junior dyads. The hypothesis of senior interaction preference receives at best weak and context-dependent support.

Finally, Hypothesis H3 tests the effect of gender-based status cues (as opposed to professional status cues as in H2) on interaction frequency. We use two sub-hypotheses to examine whether women at the junior or senior level are more likely to interact with men than other women.

For junior dyads (H3a), we find no support across the three organisational contexts. The interaction terms show varying directional evidence ($\beta_{18}^{uni}$ = 0.33, 95% CI [−0.16, 0.70], pd = 0.91; $\beta_{18}^{lab}$ = 0.46, 95% CI [0.28, 0.63], pd = 1.00; $\beta_{18}^{bus}$ = −0.00, 95% CI [−8.10, 8.08], pd = 0.50), while the combined probability ratios comparing women-women to mixed-gender junior dyads show insufficient evidence in all contexts (PR$^{uni}$ = 0.98, 95% CI [0.32, 2.22], pd = 0.52; PR$^{lab}$ = 0.84, 95% CI [0.57, 1.23], pd = 0.81; PR$^{bus}$ = 1.30, 95% CI [0.00, 1.6e + 07], pd = 0.51). All CI include 1 and none provide meaningful directional evidence for H3a. The organisational contrast between universities and laboratories shows insufficient evidence of a substantial difference ($\Delta\beta_{18}^{lab-uni}$ = 0.13, 95% CI [−0.34, 0.64], pd = 0.69).

For senior dyads (H3b), we find strongly context dependent patterns. Universities demonstrate insufficient evidence ($\beta_{16}^{uni}$ = −0.05, 95% CI [−0.88, 0.60], pd = 0.56; PR$^{uni}$ = 0.44, 95% CI [0.09, 1.54], pd = 0.89). Research labs provide strong support for H3b, with women-women senior dyads approximately half as likely to interact next as mixed-gender senior dyads ($\beta_{16}^{lab}$ = 0.22, 95% CI [−0.02, 0.45], pd = 0.96; PR$^{lab}$ = 0.52, 95% CI [0.30, 0.90], pd = 0.99). In stark contrast, private companies show strong evidence against H3b, with women-women senior dyads approximately 25 times as likely to interact next ($\beta_{16}^{bus}$ = 1.43, 95% CI [0.65, 2.30], pd = 1.00; PR$^{bus}$ = 25.00, 95% CI [1.58, 200], pd = 0.99), though the magnitude remains highly uncertain, due to the exceptionally wide CI, which ranges from 1.58 to 200.

Overall, H3a receives no support, while H3b shows opposite patterns in research labs versus private companies ($\Delta\beta_{16}^{bus-lab}$ = 1.21, 95% CI [0.39, 2.12], pd = 1.00). This dramatic context-dependency suggests that gender-seniority interactions are fundamentally shaped by organisational characteristics—potentially including gender representation, hierarchical structures, or workplace culture—rather than reflecting universal interaction patterns.

Beyond the three focal hypotheses, our model includes additional exogenous (friendship, advice-seeking, co-location, age, tenure) and endogenous effects (inertia, indegree, recency). Results for these effects should be considered exploratory, as they are not linked to specific theoretical frameworks nor prior evidence.

We find strong evidence that *friendship ties* increase interaction probability in universities and private companies ($\beta_{6}^{uni}$ = 0.71, 95% CI [0.27, 1.18], pd = 1.00; $\beta_{6}^{bus}$ = 1.01, 95% CI [0.47, 1.54], pd = 1.00), but there is inconclusive directional evidence for research labs ($\beta_{6}^{lab}$ = 0.00, 95% CI [−0.19, 0.19], pd = 0.51).

For *advice-seeking*, we find evidence of positive effects across all contexts, strongest in private companies ($\beta_{7}^{bus}$ = 1.52, 95% CI [0.95, 2.15], pd = 1.00), followed by research labs ($\beta_{7}^{lab}$ = 0.30 (95% CI [0.16, 0.43], pd = 1.00), with weak evidence in universities ($\beta_{7}^{uni}$ = 0.20 (95% CI [−0.06, 0.46], pd = 0.93).

*Co-location* shows weak positive effects in academic contexts ($\beta_{8}^{uni}$ = 0.15, 95% CI [−0.01, 0.32], pd = 0.97; $\beta_{8}^{lab}$ = 0.09, 95% CI [−0.02, 0.20], pd = 0.95) but insufficient evidence in private companies ($\beta_{8}^{bus}$ = −0.35, 95% CI [−0.96, 0.28], pd = 0.86). All credible intervals include zero, reflecting residual uncertainty about effect magnitude.

*Age-based homophily*, defined as team members that pertain to the same age category, demonstrates strong evidence only in research labs ($\beta_{19}^{lab}$ = 0.10, 95% CI [0.01, 0.19], pd = 0.99), with insufficient evidence of directionality in both the universities and private companies ($\beta_{19}^{uni}$ = −0.10, 95% CI [−0.27, 0.07], pd = 0.87; $\beta_{19}^{bus}$ = −0.02, 95% CI [−0.23, 0.21], pd = 0.57).

*Tenure homophily*, when both actors are in the same tenure category, shows no clear effects in any context ($\beta_{20}^{uni}$ = −0.07, 95% CI [−0.20, 0.05], pd = 0.88; $\beta_{20}^{lab}$ = 0.05, 95% CI [−0.05, 0.16], pd = 0.86; $\beta_{20}^{bus}$ = −0.22, 95% CI [−0.54, 0.09], pd = 0.91).

*Inertia* assesses whether the occurrence of past interactions increases the probability of future interactions. We find evidence of a negative effects across all contexts ($\gamma_{1}^{uni}$ = −0.52, 95% CI [−0.92, −0.15], pd = 0.99; $\gamma_{1}^{lab}$ = −0.37, 95% CI

[−0.79, 0.05], pd = 0.96; $\gamma_1^{bus}$ = −0.65, 95% CI [−1.12, −0.08], pd = 0.99), suggesting past interactions do not increase the probability of future interactions. The random intercepts for inertia show modest variation across teams (σ = 0.14, 95% CI [0.05, 0.32], pd = 1.00), with most team-specific deviations remaining close to zero. University teams show the widest spread, ranging from substantial negative effects (Team 3: γ = −0.43, 95% CI [−0.99, −0.01], pd = 0.98) to weak positive tendencies (Team 9: γ = 0.30, 95% CI [−0.03, 0.71], pd = 0.96), though most credible intervals include zero. Research lab and private company teams cluster more tightly around zero, showing minimal between-team variation in inertia effects. The limited heterogeneity suggests that while some organisational context matters (as reflected in fixed effects), within-organisation variation across teams is relatively small for this effect.

*Indegree* in the network of past events serves as an indicator of an actor's centrality, assessing whether central actors are likely to be part of future events. We find strong positive effects in universities and research labs ($\beta_2^{uni}$ = 1.30, 95% CI [0.84, 1.76], pd = 1.00; $\beta_2^{lab}$ = 1.05, 95% CI [0.73, 1.37], pd = 1.00), indicating central actors are more likely to be involved in future interactions, but insufficient evidence in private companies ($\beta_2^{bus}$ = 0.49, 95% CI [−0.40, 1.37], pd = 0.86).

*Recency* captures the extent to which two consecutive interactions occur between the same dyad. We find a strong positive effect across all contexts ($\beta_3^{uni}$ = 1.41, 95% CI [1.24, 1.58], pd = 1.00; $\beta_3^{lab}$ = 1.10, 95% CI [0.99, 1.20], pd = 1.00; $\beta_3^{bus}$ = 1.99, 95% CI [1.66, 2.32], pd = 1.00). The next interaction is most likely to occur between the same dyad as the most recent one, consistent with face-to-face interactions remaining stable for periods (e.g., during meetings). This strong recency effect does not reflect sensor artefacts, as consecutive detections are aggregated into single interactions.

## Discussion

This article makes several contributions to the literature on team diversity. First, we demonstrate the feasibility of analysing high-resolution interaction data collected via wearable sensors using a Dynamic Actor Network Model (DyNAM). By constructing dyadic configurations based on gender and seniority, we show how interaction frequencies vary across pairings and organisational contexts. Although results do not consistently support our hypotheses, they reveal that gender homophily and status effects differ by context and seniority level, underscoring the need to move beyond aggregate team-level measures such as overall gender ratios. Aggregate team composition constructs offer only a coarse approximation compared to the nuanced perceptions and interaction choices of individual members, which may help explain inconsistent findings on gender diversity and performance [2,3]. Similarly, applying DyNAM illustrates the value of a time-sensitive analysis of interaction behaviour, providing an alternative to reliance on self-reports [7]. To our knowledge, our study is among the first to apply a hierarchical DyNAM to real-world interaction data across 11 R&D teams. Importantly, the absence of uniform results highlights the complexity of face-to-face interaction and cautions against simplistic, linear assumptions about diversity effects. A microdynamic perspective thus offers a more realistic foundation for future research on diversity in teams.

Although this study does not provide clear-cut evidence across all teams and organisational contexts, the absence of gender homophily is relatively consistent: mixed-gender dyads show higher interaction likelihood in universities, research labs, and private companies. Where evidence exists, the similar-attraction paradigm does not hold, and gender-heterophily appears to drive interaction patterns—a positive finding for diversity, as it prevents restricted access to varied competencies and information. Interestingly, in the university context, the point estimate for men-only dyads suggests a 1.66-fold increase in interaction likelihood. While inconclusive, this may reflect the more purpose-driven nature of university interactions, where private offices and teaching duties limit chance encounters, prompting men to cluster. Physical co-location could be key to explaining differences in gender homophily among men versus women, as noted by Yang et al. [151].

Our findings underscore the importance of organisational context in team diversity research. While this is well established in the literature [138], our behavioural analysis shows that status effects vary across contexts. In universities, senior members interact more frequently than juniors, whereas in private companies, junior dyads dominate. This likely reflects the hierarchical structure of university teams, which span multiple career stages—from entry-level to full

professors—compared to the flatter, more horizontal structure of private companies. In shared office environments like those observed in the private company, such flat structures offer insufficient hierarchical differentiation to identify central actors by seniority. Supporting this interpretation, research labs—whose hierarchies resemble those of universities—show a point estimate indicating senior dyads are 1.16 times more likely to interact. Although not conclusive, these results suggest that in organisational contexts with clearly defined hierarchies, senior members tend to occupy more central positions, a pattern absent in flatter organisational settings.

Similarly, the importance of organisational context becomes evident when examining Hypothesis 3 on gender-based status effects. Notably, point estimates reveal a marked asymmetry: interaction among women-only dyads is more likely in private companies but less likely in universities and research labs. Although evidence remains inconclusive, this divergence underscores the relevance of gendered status dynamics. In university and lab settings, women within the same seniority level tend to interact with men—the higher gender-status group—whereas in private companies, women more often interact with other women of similar seniority.

One possible explanation of this divergence is the persistence of the "chilly climate" [152], where overt and subtle forms of discrimination [153] create an unwelcoming environment for women especially in the private company context. As a result, women may interact primarily with others of the same gender within their seniority level, likely as a form of mutual support in a demanding context [154]. This tendency is particularly pronounced among senior women, reflecting the "gender-status mismatch", where high professional status clashes with low gender-based status, prompting avoidance of interactions that challenge stereotypes. Given that this pattern is primarily observed in the business environment as opposed to universities or research labs suggests that the chilly climate is stronger in the private company. Unlike university or laboratory settings, where career paths are governed by explicit accreditation standards, private firms often feature flat, informal, yet intensely competitive work environments. These less regulated environments tend to be more chilly for women, as the lack of formal benchmarks heighten direct, male-oriented competition and exclusion. However, flat organisational structures in business rely at the same time more heavily on gender-based status cues, which serve as important interaction signals in the absence of formal coordination mechanisms. As gender-status cues exert greater influence, the "gender-status mismatch" [98] emerges as a compelling alternative explanation for women-only interaction. As argued, especially senior women face a deficit of trust within their professional networks, where their work-related high status is incongruent with their gender-based low status. Being perceived as a status threat by male colleagues, mixed-gender interactions become thus less likely. Therefore, when senior women dyads interact more frequently, it potentially is as much about avoiding the pitfalls of mixed-gender status threats as it is about seeking mutual support in chilly business environments.

This study has several limitations. First, data collection was limited to one week per team. Although field observations and interviews suggest that this period reflects a typical week—including team meetings and member availability—it remains unclear whether interaction patterns are stable across different time scales such as days, weeks, or months. Future research could examine how these patterns evolve over time, whether they exhibit specific temporal dynamics, and how such dynamics influence outcomes. Recent work [155], for example, shows that the speed of tie formation within networks affects relational outcomes and network returns.

Second, any interpretation of what these interaction patterns imply for team members remains necessarily speculative. Although we conducted qualitative interviews with at least three members per team, these insights do not easily scale to a hierarchical model of team interaction where dyadic effects aggregate across teams. Future studies could adopt a mixed-methods approach, combining DyNAM models of observed interaction patterns with qualitative accounts of their perceived significance.

Finally, the decision to discretise continuous variables such as team tenure and age was driven by the analytical requirements of our model. Using continuous measures complicates interpretation of log-likelihood differences. Similarly, team roles were aggregated into binary categories (*Senior* vs. *Junior*). Although descriptive statistics in the supplementary

S1 File–Table 1 provide some insight into variable distributions—partially explaining the wide credible intervals in private companies—alternative categorisations might yield different results.

## Supporting information

**S1 File. Supplementary tables including descriptive statistics of socio-demographic variables and interaction statistics.**
(DOCX)

**S2 File. Supplementary tables including DyNAM model statistics.**
(DOCX)

## Acknowledgments

We would like to thank the anonymous reviewers for their very helpful and constructive comments. We would also like to thank the members of the Stadtfeld Group at the ETH Zürich Social Networks Lab for initial discussions and guidance.

## Author contributions

**Conceptualization:** Jörg Müller.

**Data curation:** Jörg Müller.

**Formal analysis:** Alvaro Uzaheta.

**Funding acquisition:** Jörg Müller.

**Methodology:** Jörg Müller, Alvaro Uzaheta, Julián Salas Piñón.

**Project administration:** Jörg Müller.

**Software:** Alvaro Uzaheta, Julián Salas Piñón.

**Writing – original draft:** Jörg Müller, Alvaro Uzaheta.

**Writing – review & editing:** Jörg Müller, Alvaro Uzaheta, Julián Salas Piñón.

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
