## [Decision Letter · Decision Letter 0]

28 May 2025

PONE-D-25-13840Gender, Status, and Team Interaction: A Microdynamic Exploration of Wearable Sensor Data Across 11 Research Groups.PLOS ONE

Dear Dr. Muller,

Thank you for submitting your manuscript to PLOS ONE. After careful consideration, we feel that it has merit but does not fully meet PLOS ONE’s publication criteria as it currently stands. Therefore, we invite you to submit a revised version of the manuscript that addresses the points raised during the review process.

Both reviewers commend the manuscript’s innovative use of hierarchical DyNAM on fine-grained wearable-sensor data, but they converge on several critical revisions needed before the work can meet PLOS ONE’s methodological and reporting standards. I will briefly summarise and not go into much detail here as the reviewers have done a great job in providing detailed, constructive action points for the authors:

The theoretical framing must engage more fully with the dynamic-network and relational-event literature (e.g., Pilny, Contractor, Bianchi et al., as suggested by Reviewer 1) and streamline the arguments to outline a tighter set of hypotheses that are clearly anchored in information-processing and theory on homophily.Arguably the most important point for revision, is the concerns about the treatment of heterogeneous samples raised by Reviewer 1. The authors must justify the treatment heterogeneous, cross-national sample, the split of “Team 7,”, and present clear evidence for why they treat the data this way (either through e.g., descriptives, sensitivity analyses). The authors should also provide further justification of their treatment of variables, such as binning of continuous covariates, and supply basic descriptives (team size, gender, seniority) to allow readers to assess comparability.The methodological section should include an outline of the DyNAM equation implemented in the manuscript, an explicit definition of the “microdynamic perspective,” and transparent criteria for interpreting Bayesian output; statistical reporting should be complete and consistent across all effects.Finally, try to ensure that figures, tables, and variable labels are self-explanatory, and thoroughly proof read to ensure that the writing and grammar are as clear and consistent as possible (e.g., no missing spaces, contractions, ambiguous terms such as “significant others”).

Please attend to these key points, as well as all other reviewer feedback in your revision.

We look forward to receiving your revised manuscript.

Kind regards,

Daniel Redhead

Academic Editor

PLOS ONE

Journal Requirements:

2. Thank you for stating the following financial disclosure: [This work has received funding from the European Union’s Horizon 2020 research and innovaton programme under grant agreement No 665851.].

**Comments from PLOS Editorial Office:** We note that one or more reviewers has recommended that you cite specific previously published works. As always, we recommend that you please review and evaluate the requested works to determine whether they are relevant and should be cited. It is not a requirement to cite these works. We appreciate your attention to this request.

Reviewers' comments:

Reviewer's Responses to Questions

**Comments to the Author**

1. Is the manuscript technically sound, and do the data support the conclusions?

Reviewer #1: Partly

Reviewer #2: Partly

2. Has the statistical analysis been performed appropriately and rigorously? 

Reviewer #1: Yes

Reviewer #2: I Don't Know

3. Have the authors made all data underlying the findings in their manuscript fully available?

Reviewer #1: No

Reviewer #2: Yes

4. Is the manuscript presented in an intelligible fashion and written in standard English?

Reviewer #1: Yes

Reviewer #2: Yes

5. Review Comments to the Author

Reviewer #1: This manuscript presents a methodologically interesting case study applying a hierarchical Dynamic Actor Network Model (DyNAM) to wearable sensor data capturing face-to-face interactions within R&D teams. The empirical setting of exploring team dynamics using such granular, real-world data is relevant and aligns well with PLOS ONE's interest in methodologically rigorous research. The application of DyNAM to this specific context also appears correct and represents a valuable extension of this method to new empirical settings, particularly within organizational studies.

Several areas require significant attention to strengthen the theoretical framing, methodological reporting, data justification, and overall presentation for publication in PLOS ONE.

1. Literature Review and Theoretical Framing

o While the paper correctly identifies the shift towards a dynamic understanding of team processes and appropriately uses a DyNAM, the abstract and likely the full paper could benefit from a more comprehensive engagement with the broader literature on using social network models, particularly event-based models like Relational Event Models (REMs) and actor-based models, to study organizational settings. Key works by researchers such as Pilny, Lendeers, and Contractor – just to name a few – have significantly contributed to this area and should be referenced to properly contextualize the study within the existing field of organizational studies using dynamic network models.

o It is also recommended that the authors include a reference to Bianchi et al.’s (2024) recent review on Relational Event Modeling. This review provides an overview of recent methodological and empirical developments in event-based network analysis, which is directly relevant to the paper's chosen method. Citing this review will also provide readers with valuable guidance on the state of the art in this domain.

o The paper presents a substantial number of hypotheses (10, as noted). While exploring multiple facets of interaction is valuable, presenting this many distinct hypotheses in a single paper can sometimes dilute the core theoretical argument. The authors should review and potentially consolidate the hypotheses, ensuring each is clearly derived from a well-articulated theoretical argument and connected explicitly to the existing literature on gender, status, and team interaction dynamics.

2. Dataset and Sampling Approach

o A significant concern arises from the heterogeneity of the dataset used in the analysis. Combining data from R&D teams collected in different years (2016/2017 vs. 2018), across different countries (UK and Spain), from varied organizational sectors (private vs. public), different environments (university, research labs, private company), and diverse scientific fields (biomedical engineering, construction, nanoscience, energy engineering) raises questions about the comparability of these groups.

o Specifically, the perception and influence of "status" can vary significantly across national cultures (UK vs. Spain), organizational types (university vs. private company), and professional fields. Aggregating data without clearly accounting for these potential sources of variation might obscure or misinterpret the actual dynamics at play.

o The splitting of "Team 7" into sections A and B also requires clear justification. Is this split based on a meaningful organizational or interactional boundary within that group, or is it an artifact of data collection or analytical choices? The rationale for this split and how data from these two sections are treated in the hierarchical model should be explicitly explained.

3. Methodological Reporting

o While the abstract mentions a hierarchical DyNAM, the full paper appears to lack a clear, explicit mathematical equation specifying the model used. For a complex statistical model like DyNAM, a formal equation is crucial for clarity, reproducibility, and for reviewers and readers to fully understand how the different effects (gender, status, homophily, team-level variations) are modeled. The current approach of inferring the model specification from hypotheses and figures is insufficient.

o The term "microdynamic perspective" is used in the abstract and likely in the paper. While the intuitive meaning – dynamic interactions at the individual/dyadic level and their emergent patterns – is understandable, providing a more explicit definition grounded in the literature would be beneficial. Clarifying that this perspective emphasizes how higher-level network configurations or team processes emerge from the aggregation of local, time-varying interactions would enhance conceptual clarity.

4. Writing and Formatting

o Attention to detail in writing and formatting is important for a professional publication. On page 9 ("status expectations...."), missing spaces around inline mathematical notation need to be corrected.

o The use of contractions (e.g., "It's" on page 17) should be reviewed and made consistent with the overall writing style of the manuscript.

In conclusion, the paper presents a valuable application of dynamic network modeling to a relevant topic, and addressing the identified points is crucial to enhance its quality and impact.

Reviewer #2: Thank you for the opportunity to review this manuscript on how gender and status affect interaction in teams. This manuscript offers an innovative application of behavioural measurements that offers great insight into team dynamics and that the field can benefit from. However, I do have some concerns regarding the current manuscripts that I will detail below.

Introduction

• There is a lot of jumping between theory/ empirical evidence and the current manuscript. This should be streamlined more to make the manuscript more precise in language and more stringent in its argumentative structure. For example, first, after the general introduction, describe the theoretical background and empirical evidence (microdynamic perspectives of gender and status diversity in groups), then the methodological background (focus on behavioural data social network data), then the current manuscript that combines both (behavioural data via wearable sensors, establishing DyNAM to analyse social network data and using both to test hypotheses according to microdynamic perspectives on team diversity).

• Based on the current theoretical background, the incorporation of informal social ties into the hypothesis needs stronger basis (see my comment regarding hypothesis H1 and H1b).

Hypotheses:

• Hypotheses 1a and 1b are not straightforwardly derived from the information processing perspective. While authors do describe their reasoning, it should be made clearer that interaction frequency acts as a foundation for information exchange and thus deeper information processing. That increased interaction frequency leads to deeper information processing is an auxiliary hypothesis that is not tested. Instead, the hypotheses postulate that social ties affect interaction frequencies. The only basis for these hypotheses is evidence regarding social ties and team performance, so currently, the hypotheses seem misplaced when considering them from the information processing perspective. The arguments (empirical or theoretical) for Hypothesis 1a and H1b need more strengthening.

• While authors acknowledge that Hypothesis 3d can be explained by homophily, they neglect to mention that the same is the case for Hypothesis 3a.

Method

• The total number of teams should be described first, then how they were collected. The sentence at the end of the paragraph can simply be moved up.

• The mean and standard deviation of group sizes should also be reported, as well as a broader description of the sample (age, gender distribution, senior-junior distribution). Currently, the sample description is mostly missing. Differences in e.g. gender composition in teams might affect how results are or can be interpreted.

• After reading the results part, it appears to me that the decision to bin age and team tenure was made due to analytical demands of the DyNAM. I do not agree that (generally speaking) categorizing continuous variables would improve interpretability as it might muddle effects for individuals close to cuff-off values. As the authors report no information on the descriptive distribution of their variables, it is impossible to judge how this categorization might affect the reported results. This should be discussed as a limitation.

• The authors only mention team tenure in the methods and results but have not made /described any specific assumptions regarding team tenure. Results reported should thus be marked as exploratory. The same is the case for results regarding the endogenous effects.

Results

• In the first sentence of the results (page 18), the authors refer to a “sub model”. It is unclear what this sub-model refers to.

• The authors are not consistent in their reports of test statistics. For some results, they report all necessary test statistics for the effect parameters, for some they only report the test statistics for the probability ratio. In the latter cases, test statistics for the effect parameters should also be reported and interpreted.

• In general, the inference criteria on which the authors base their judgements regarding the evidence (e.g., what consists of strong evidence) remains unclear. This makes it difficult to follow author judgements regarding how strong the support for the hypotheses is. It is also unclear how (or whether) the combination of test statistics is used to come to conclusions regarding evidence in support for the hypothesis. The authors report test statistics for choice probability (mean, CI and posterior probability) as well as test statistics of the probability ratio (incl. CI and posterior probability). In the methods section, authors described that the parameters of DyNam are interpreted regarding the direction and statistical significance. In Bayesian analyses, this “statistical significance” should be further defined. The authors refer to the credible interval for their interpretation, but in general understanding, CIs that include zero should be interpreted as not significant. This is the case for e.g., the parameter for age category which the authors do not report. Regarding the probability ratio, the authors describe that it can be interpreted like Odds Ratio, meaning the CI should not include one to assume statistical significance in the frequentist framework. Yet, this is the case for two results for which the authors judge support for their hypothesis (age category and advice relationships among same status same-sex men). The authors should clearly describe the criteria by which the interpret results as supporting their assumptions.

• Without reading the text, it is not possible to understand figures and tables – the variable names are ambiguous. Additionally, it is unclear what “random effects” figure 2 refers to.

Discussion

• The summary of results at the beginning of the discussion is not differentiated enough to reflect interpretation of different levels of evidence in support of the hypotheses.

• The word choice is “significant others” is misguiding as this term is commonly used to refer to romantic partners. I would recommend using a different term.

Details:

• grammar – “DyNam also considers the activity…” (Page 15)

• inconsistent time use page 17

6. PLOS authors have the option to publish the peer review history of their article (what does this mean?). If published, this will include your full peer review and any attached files.

Reviewer #1: No

Reviewer #2: No

---

## [Author Response · Author response to Decision Letter 1]

28 Nov 2025

We copy here our detailed reply to the editors comments and reviewers feedback. These have been provided as a formatted word document for easier readability.

We would like to express our gratitude to the anonymous reviewers for their close reading and very helpful critique. In what follows, we describe how each point has been addressed.

Editors feedback:

[1] The theoretical framing must engage more fully with the dynamic-network and relational-event literature (e.g., Pilny, Contractor, Bianchi et al., as suggested by Reviewer 1) and streamline the arguments to outline a tighter set of hypotheses that are clearly anchored in information-processing and theory on homophily.

REPLY: We have incorporated the literate on REM and DyNAM in a dedicated section after the conceptual framework. We also have streamlined the argument regarding gender and status effects in teams within the diversity literature. This has led us to formulate three hypothesis: 1) gender-homophily, 2) status-effects, 3) gender-based status effects.

[2] Arguably the most important point for revision, is the concerns about the treatment of heterogeneous samples raised by Reviewer 1. The authors must justify the treatment heterogeneous, cross-national sample, the split of “Team 7,”, and present clear evidence for why they treat the data this way (either through e.g., descriptives, sensitivity analyses). The authors should also provide further justification of their treatment of variables, such as binning of continuous covariates, and supply basic descriptives (team size, gender, seniority) to allow readers to assess comparability.

REPLY: We have incorporated a clear justification for Team 7 (same group but different locations) and argued for weak influence of national context. We agree on the importance of the organisational context, however. This has led us to present the analysis for each of the organisational contexts separately: university, research lab, and private company. We have also included in-depth descriptive analysis of team characteristics, including network visualization of interactions, and descriptive statistics on main socio-demographic variables used.

[3] The methodological section should include an outline of the DyNAM equation implemented in the manuscript, an explicit definition of the “microdynamic perspective,” and transparent criteria for interpreting Bayesian output; statistical reporting should be complete and consistent across all effects.

REPLY: we have included the DyNAM equation implemented in the methods section and expanded on our definition of microdynamic perspective. We now also consistently report the posterior median as the point estimate for central tendency and the 95% central credible interval (CI) to quantify uncertainty for all model parameters (β) and derived measures.

Interpretation Criteria: We have added an explicit statement in the methods section defining our interpretation criteria: we consider there to be strong statistical evidence for an effect if the 95% CI strictly excludes zero (for raw parameters) or one (for probability ratios). We have revised the methodology and results sections to establish a standardized reporting framework.

[4] Finally, try to ensure that figures, tables, and variable labels are self-explanatory, and thoroughly proof read to ensure that the writing and grammar are as clear and consistent as possible (e.g., no missing spaces, contractions, ambiguous terms such as “significant others”).

REPLY: we have adapted the figures and tables captions, including variable labels. Proof reading has been carried out, using AI-based tools.

Reviewer #1: This manuscript presents a methodologically interesting case study applying a hierarchical Dynamic Actor Network Model (DyNAM) to wearable sensor data capturing face-to-face interactions within R&D teams. The empirical setting of exploring team dynamics using such granular, real-world data is relevant and aligns well with PLOS ONE's interest in methodologically rigorous research. The application of DyNAM to this specific context also appears correct and represents a valuable extension of this method to new empirical settings, particularly within organizational studies.

Several areas require significant attention to strengthen the theoretical framing, methodological reporting, data justification, and overall presentation for publication in PLOS ONE.

1. Literature Review and Theoretical Framing

• While the paper correctly identifies the shift towards a dynamic understanding of team processes and appropriately uses a DyNAM, the abstract and likely the full paper could benefit from a more comprehensive engagement with the broader literature on using social network models, particularly event-based models like Relational Event Models (REMs) and actor-based models, to study organizational settings. Key works by researchers such as Pilny, Lendeers, and Contractor – just to name a few – have significantly contributed to this area and should be referenced to properly contextualize the study within the existing field of organizational studies using dynamic network models.

• It is also recommended that the authors include a reference to Bianchi et al.’s (2024) recent review on Relational Event Modeling. This review provides an overview of recent methodological and empirical developments in event-based network analysis, which is directly relevant to the paper's chosen method. Citing this review will also provide readers with valuable guidance on the state of the art in this domain.

REPLY: we have included the most relevant REM/DyNAM literature in a dedicate section. Although we had an additional section summarising the methodological extensions of the REM model, this is not the focus of this article. We hope that the provide section clarifies the approach used sufficiently.

• The paper presents a substantial number of hypotheses (10, as noted). While exploring multiple facets of interaction is valuable, presenting this many distinct hypotheses in a single paper can sometimes dilute the core theoretical argument. The authors should review and potentially consolidate the hypotheses, ensuring each is clearly derived from a well-articulated theoretical argument and connected explicitly to the existing literature on gender, status, and team interaction dynamics.

REPLY: we have streamlined our argument and reduced the original 10 hypotheses to three main hypotheses. However, the use of DyNAM still requires to operationalise these three hypotheses as two sub-hypothesis each: 3 x 2 = 6.

We have also revised the role of advice and friendship ties previously associated to the information processing perspective. This has been removed, as we don’t have behavioural evidence in our data. We still introduce the information processing perspective as part of the overall conceptual framework but point out that this does not yield testable hypothesis in our approach.

2. Dataset and Sampling Approach

• A significant concern arises from the heterogeneity of the dataset used in the analysis. Combining data from R&D teams collected in different years (2016/2017 vs. 2018), across different countries (UK and Spain), from varied organizational sectors (private vs. public), different environments (university, research labs, private company), and diverse scientific fields (biomedical engineering, construction, nanoscience, energy engineering) raises questions about the comparability of these groups.

• Specifically, the perception and influence of "status" can vary significantly across national cultures (UK vs. Spain), organizational types (university vs. private company), and professional fields. Aggregating data without clearly accounting for these potential sources of variation might obscure or misinterpret the actual dynamics at play.

• The splitting of "Team 7" into sections A and B also requires clear justification. Is this split based on a meaningful organizational or interactional boundary within that group, or is it an artifact of data collection or analytical choices? The rationale for this split and how data from these two sections are treated in the hierarchical model should be explicitly explained.

REPLY: We argue in more detail for the heterogenous sample, clarifying that Team 7 is located in two distant cities. We also cite evidence that national culture has weak influence on organisational processes. We agree with the importance of organisational contexts and have split our results and reporting according to university, research lab and private company context.

3. Methodological Reporting

• While the abstract mentions a hierarchical DyNAM, the full paper appears to lack a clear, explicit mathematical equation specifying the model used. For a complex statistical model like DyNAM, a formal equation is crucial for clarity, reproducibility, and for reviewers and readers to fully understand how the different effects (gender, status, homophily, team-level variations) are modeled. The current approach of inferring the model specification from hypotheses and figures is insufficient.

REPLY: we have inserted the mathematical equation for our model in the methods section.

• The term "microdynamic perspective" is used in the abstract and likely in the paper. While the intuitive meaning – dynamic interactions at the individual/dyadic level and their emergent patterns – is understandable, providing a more explicit definition grounded in the literature would be beneficial. Clarifying that this perspective emphasizes how higher-level network configurations or team processes emerge from the aggregation of local, time-varying interactions would enhance conceptual clarity.

REPLY: we have introduced a more precise definition of what we mean by a microdynamic perspective at the start of the section “Analytical approach”. We hope that this definition is sufficiently precise.

4. Writing and Formatting

• Attention to detail in writing and formatting is important for a professional publication. On page 9 ("status expectations...."), missing spaces around inline mathematical notation need to be corrected.

• The use of contractions (e.g., "It's" on page 17) should be reviewed and made consistent with the overall writing style of the manuscript.

In conclusion, the paper presents a valuable application of dynamic network modeling to a relevant topic, and addressing the identified points is crucial to enhance its quality and impact.

REPLY: we have corrected the formatting issues.

Reviewer #2: Thank you for the opportunity to review this manuscript on how gender and status affect interaction in teams. This manuscript offers an innovative application of behavioural measurements that offers great insight into team dynamics and that the field can benefit from. However, I do have some concerns regarding the current manuscripts that I will detail below.

Introduction

• There is a lot of jumping between theory/ empirical evidence and the current manuscript. This should be streamlined more to make the manuscript more precise in language and more stringent in its argumentative structure. New structure:

o For example, first, after the general introduction, describe the theoretical background and empirical evidence (microdynamic perspectives of gender and status diversity in groups),

o methodological background (focus on behavioural data social network data),

o then the current manuscript that combines both (behavioural data via wearable sensors, establishing DyNAM to analyse social network data and using both to test hypotheses according to microdynamic perspectives on team diversity).

REPLY: we have streamlined our argument, by developing first the conceptual framework on gender and status and team interaction. We then outline the REM/DyNAM approach in a dedicated section describing our analytical approach before presenting the hypothesis that result from combining these two approaches. We hope this provides a more concise argument.

• Based on the current theoretical background, the incorporation of informal social ties into the hypothesis needs stronger basis (see my comment regarding hypothesis H1 and H1b).

REPLY: we have simplified our argument and remove advice-seeking and friendship ties from our hypothesis. We still report the effects for instrumental and expressive ties as part of the DyNAM model, but do not examine these in connection with specific hypothesis. This could be explored in a future publication, examining especially how friendship and advice seeking ties change in relation to gender and seniority.

Hypotheses:

• Hypotheses 1a and 1b are not straightforwardly derived from the information processing perspective. While authors do describe their reasoning, it should be made clearer that interaction frequency acts as a foundation for information exchange and thus deeper information processing. That increased interaction frequency leads to deeper information processing is an auxiliary hypothesis that is not tested. Instead, the hypotheses postulate that social ties affect interaction frequencies. The only basis for these hypotheses is evidence regarding social ties and team performance, so currently, the hypotheses seem misplaced when considering them from the information processing perspective. The arguments (empirical or theoretical) for Hypothesis 1a and H1b need more strengthening.

• While authors acknowledge that Hypothesis 3d can be explained by homophily, they neglect to mention that the same is the case for Hypothesis 3a.

REPLY: we have removed hypothesis H1a/b on the role of social ties (friendship and advice), as we agree with Reviewer 2 that this is not possible. We acknowledge that difficulty in the conceptual framework when describing the information processing approach.

Method

• The total number of teams should be described first, then how they were collected. The sentence at the end of the paragraph can simply be moved up.

• The mean and standard deviation of group sizes should also be reported, as well as a broader description of the sample (age, gender distribution, senior-junior distribution). Currently, the sample description is mostly missing. Differences in e.g. gender composition in teams might affect how results are or can be interpreted.

• After reading the results part, it appears to me that the decision to bin age and team tenure was made due to analytical demands of the DyNAM. I do not agree that (generally speaking) categorizing continuous variables would improve interpretability as it might muddle effects for individuals close to cuff-off values. As the authors report no information on the descriptive distribution of their variables, it is impossible to judge how this categorization might affect the reported results. This should be discussed as a limitation.

• The authors only mention team tenure in the methods and results but have not made /described any specific assumptions regarding team tenure. Results reported should thus be marked as exploratory. The same is the case for results regarding the endogenous effects.

REPLY: we provide a more detailed description of teams, including descriptive statistics of main socio-demographic variable as well as face-to-face interaction statistics with teams in the supplementary file S1. Interaction network visualisations have been included, visualising interactions among team members according to our main variables of interest: gender and seniority. We also discuss the binning of age and tenure into categorical variables as an limitation at the end of our paper. Alternative explanation involving co-location, tenure, age, friendship and advice are all marked as exploratory.

Results

• In the first sentence of the results (page 18), the authors refer to a “sub model”. It is unclear what this sub-model refers to.

• The authors are not consistent in their reports of test statistics. For some results, they report all necessary test statistics for the effect parameters, for some they only report the test statistics for the probability ratio. In the latter cases, test statistics for the effect parameters should also be reported and interpreted.

• In general, the inference criteria on which the authors base their judgements regarding the evide

---

## [Decision Letter · Decision Letter 1]

15 Jan 2026

PONE-D-25-13840R1Gender, Status, and Team Interaction: A Microdynamic Exploration of Wearable Sensor Data Across 11 Research Groups.PLOS One

Dear Dr. Muller,

Thank you for submitting your manuscript to PLOS ONE. After careful consideration, we feel that it has merit but does not fully meet PLOS ONE’s publication criteria as it currently stands. Therefore, we invite you to submit a revised version of the manuscript that addresses the points raised during the review process.

We look forward to receiving your revised manuscript.

Kind regards,

Daniel Redhead

Academic Editor

PLOS One

Journal Requirements:

Additional Editor Comments (if provided):

Reviewer 1 raises several important issues related to the interpretation of the models and suggests some good solutions. In your revision, please try to incorporate their feedback such that the interpretation better aligns with the Bayesian modelling framework that was implemented in the manuscript.

Reviewers' comments:

Reviewer's Responses to Questions

**Comments to the Author**

1. If the authors have adequately addressed your comments raised in a previous round of review and you feel that this manuscript is now acceptable for publication, you may indicate that here to bypass the “Comments to the Author” section, enter your conflict of interest statement in the “Confidential to Editor” section, and submit your "Accept" recommendation.

Reviewer #1: (No Response)

2. Is the manuscript technically sound, and do the data support the conclusions?

Reviewer #1: Partly

3. Has the statistical analysis been performed appropriately and rigorously? 

Reviewer #1: N/A

4. Have the authors made all data underlying the findings in their manuscript fully available?

Reviewer #1: Yes

5. Is the manuscript presented in an intelligible fashion and written in standard English?

Reviewer #1: Yes

6. Review Comments to the Author

Reviewer #1: Please see the attached file for my detailed review and specific comments regarding the Bayesian interpretation and the structural analysis.

Summary of Review (contained in attachment): The analytical engine (DyNAM + Bayesian inference) applied in this manuscript is powerful, but the current interpretation limits its potential. The analysis relies on a binary "significant/not significant" heuristic that discards the probabilistic value of the posterior distributions. The authors must shift from this binary testing mindset to a probabilistic one (e.g., using the Probability of Direction). Additionally, the variation across organizational contexts should be treated as a finding to be modeled rather than a contradiction.

7. PLOS authors have the option to publish the peer review history of their article (what does this mean?). If published, this will include your full peer review and any attached files.

Reviewer #1: No

---

## [Author Response · Author response to Decision Letter 2]

2 Mar 2026

We would like to thank the anonymous reviewer for the suggested improvements. In what follows, we describe how each point has been addressed.

Reviewer feedback:

[1] Report the Probability of Direction (pd). Calculate and report the percentage of the posterior distribution that shares the same sign as the median. This provides a direct probability statement (e.g., "there is a 93% probability that gender homophily is positive") which is more intuitive and accurate than a CI-based rejection.

REPLY: results reporting includes now both CI and pd values.

“To interpret the fitted models, we summarise the posterior distributions using the median as a point estimate and the 95% equal-tailed credible interval (CI), which contains the central 95% of the posterior mass and excludes the lowest and highest 2.5%. We also report the probability of direction (pd), which quantifies directional certainty by computing the proportion of posterior samples that share the same sign as the median [149].

Hypotheses are evaluated by jointly considering the probability of direction and the credible intervals. We classify evidence strength as follows: strong directional evidence when the 95% CI excludes the reference value (zero for β coefficients, one for probability ratios), this necessarily implies pd ≥ 0.975; moderate directional evidence when pd ≥ 0.95; weak directional evidence when pd ≥ 0.90; and insufficient directional evidence when pd < 0.90. Together, these metrics indicate whether an effect is likely to exist and provide its plausible range, allowing us to assess the strength of evidence for each hypothesis.”

[2] Discuss Magnitude and Uncertainty. Instead of simply accepting or rejecting hypotheses, discuss the strength of the evidence provided by the posterior. If 90% of the posterior mass is positive, this is worth discussing as a probable effect, even if the 95% CI includes zero.

REPLY: We discuss now both magnitude and uncertainty in the methods section and propose to qualify the obtained evidence as strong, moderate or no evidence based upon a combination of CI and pd values.

For example: “]). The effect is stronger in research labs, where women-women dyads are approximately one-third as likely to interact next (β₅ˡᵃᵇ = –0.54, 95% CI [–0.75, –0.33], pd = 1.00; PRˡᵃᵇ = 0.34, 95% CI [0.22, 0.51]). Private companies show insufficient evidence of any effect (β₅ᵇᵘˢ = 0.17, 95% CI [–1.18, 1.21], pd = 0.61; PRᵇᵘˢ = 1.40, 95% CI [0.09, 11.20]).”

[3] Reformulate Conclusions. Re-evaluate the "rejected" hypotheses (e.g., H2a, H3a) using these probabilistic metrics. It is likely that the data provides stronger support for your theoretical claims than the current binary interpretation suggests.

REPLY: We have removed statements regarding a clear “acceptance” or “rejection” of stated hypothesis but discuss our findings more in alignment with Bayesian approach.

[4] Treat the variation in beta-parameters across contexts as a result in itself and use the posterior distributions to compare these contexts directly.

REPLY: We have furthermore incorporated the variation across contexts in our model and results, indicating where beta-parameters differ, the strength of the evidence (CI and pd-base).

“To assess organisational heterogeneity, we report posterior statistics for the largest pairwise difference in β coefficients between contexts (Δβ), along with 95% CIs and probability of direction. These direct comparisons quantify whether observed cross-context differences are substantive and provide their plausible range.”

For example, when reporting results: “The organisational contrast between private companies and universities shows insufficient evidence of a substantial difference (β₅bus-uni = 0.52, 95% CI [-0.86, 1.61], pd = 0.79).”

[5] Visual presentation: Ensure the figures clearly communicate the density or the spread of the posterior.

REPLY: We have revised the provide illustrations.

“Figure 1 displays half-eye plots of the posterior distribution for the PRs of gender homophily by seniority level. The upper half shows the posterior densities; the lower half displays summary statistics: the median (dot), the 50% equal-tailed credible interval (thick line), and the 95% equal-tailed credible interval (thin line). This visualization allows inspection of both the full posterior and conventional interval-based summaries.”

[6] Update the y-axis labels.

REPLY: Axis labels have been updated and include now full descriptions.

---

## [Decision Letter · Decision Letter 2]

27 Apr 2026

Gender, Status, and Team Interaction: A Microdynamic Exploration of Wearable Sensor Data Across 11 Research Groups.

PONE-D-25-13840R2

Dear Dr. Muller,

We’re pleased to inform you that your manuscript has been judged scientifically suitable for publication and will be formally accepted for publication once it meets all outstanding technical requirements.

Kind regards,

Daniel Redhead

Academic Editor

PLOS One

Additional Editor Comments (optional):

Both the reviewer and myself believe that this round of revisions has substantially improved the manuscript, and that it is now at an acceptable standard for publication. The reviewer has provided some minor and helpful suggestions that you many wish to incorporate before submitting the final version.

Reviewers' comments:

Reviewer's Responses to Questions

**Comments to the Author**

1. If the authors have adequately addressed your comments raised in a previous round of review and you feel that this manuscript is now acceptable for publication, you may indicate that here to bypass the “Comments to the Author” section, enter your conflict of interest statement in the “Confidential to Editor” section, and submit your "Accept" recommendation.

Reviewer #1: All comments have been addressed

2. Is the manuscript technically sound, and do the data support the conclusions?

Reviewer #1: Yes

3. Has the statistical analysis been performed appropriately and rigorously? 

Reviewer #1: Yes

4. Have the authors made all data underlying the findings in their manuscript fully available?

Reviewer #1: Yes

5. Is the manuscript presented in an intelligible fashion and written in standard English?

Reviewer #1: Yes

6. Review Comments to the Author

Reviewer #1: (No Response)

7. PLOS authors have the option to publish the peer review history of their article (what does this mean?). If published, this will include your full peer review and any attached files.

Reviewer #1: No

---

## [Editor Report · Acceptance letter]

PONE-D-25-13840R2

PLOS One

Dear Dr. Muller,

I'm pleased to inform you that your manuscript has been deemed suitable for publication in PLOS One. Congratulations! Your manuscript is now being handed over to our production team.

Kind regards,

on behalf of

Dr. Daniel Redhead

Academic Editor

PLOS One